# Evaluation of Safety and Probiotic Potential of *Enterococcus faecalis* MG5206 and *Enterococcus faecium* MG5232 Isolated from Kimchi, a Korean Fermented Cabbage

**DOI:** 10.3390/microorganisms10102070

**Published:** 2022-10-20

**Authors:** YongGyeong Kim, Soo-Im Choi, Yulah Jeong, Chang-Ho Kang

**Affiliations:** MEDIOGEN, Co., Ltd., Biovalley 1-ro, Jecheon-si 27159, Korea

**Keywords:** *Enterococcus*, probiotics, safety, toxicity

## Abstract

The purpose of this study was to evaluate the genotypic and phenotypic toxicity of *Enterococcus faecalis* MG5206 and *Enterococcus faecium* MG5232 isolated from kimchi (fermented vegetable cabbage). In this study, the genotypic toxicity of the strains MG5206 and MG5232 was identified through whole-genome sequencing analysis, and phenotypic virulence, such as susceptibility to antibiotics, hemolytic activity, and gelatinase and hyaluronidase activities, was also evaluated. In addition, the in vivo toxicity of both strains was evaluated using an acute oral administration test in Sprague–Dawley rats. In all the tests, both the strains were determined to be safety by confirming that they did not show antibiotic resistance or virulence factors. In addition, these strains exhibited a low level of autoaggregation ability (37.2–66.3%) and hydrophobicity, as well as a high survival rate in gastrointestinal condition in vitro. Therefore, the safety and high gastrointestinal viability of *E. faecalis* MG5206 and *E. faecium* MG5232 suggests that both the strains could be utilized in food as potential probiotics in the future.

## 1. Introduction

*Enterococcus* spp., a group of lactic acid bacteria (LAB), constitute a large proportion of the autochthonous microflora associated with the gastrointestinal tract and are capable of colonizing in foods of animal origin. More than over 50 *Enterococcus* spp. have been reported, and *E. faecalis* and *E. faecium* are the most common types of enterococci that cause major infections in humans. *Enterococcus* spp. are Gram-positive, anaerobic, facultative, catalase-negative, and non-sporulating bacteria [1]. They have a strong tolerance to salt and acid and play a beneficial role as a starter, can serve as adjunct cultures in dairy, and help to develop cheese flavors [2,3]. However, only a few species, including *E. faecium* and *E. faecalis*, are used as probiotics because of safety issues, such as antibiotic resistance and virulence [4].

*Enterococc*us spp. are known as opportunistic pathogens causing nosocomial infections, such as endocarditis, bacteremia, and urinary tract infections (UTIs) [5,6]. Some strains are resistant to antibiotics, particularly vancomycin, which increases their pathogenicity. Moreover, various virulence factors of *Enterococcus* spp., such as tissue invasion factors, cytotoxins, and aggregation substances, have been reported. Furthermore, because of the possibility of horizontal gene transfer, it is difficult to differentiate between pathogenic and non-pathogenic *Enterococcus* spp. [7,8,9]. Therefore, since *Enterococcus* spp. are likely to be highly intrinsically virulent, it is necessary to ensure that it is a safe strain before using it in the industry.

Kimchi, a traditional Korean fermented vegetable cabbage, is commonly regarded as a health-enhancing functional food because of its diverse bioactive components, including vitamins, flavonoids, polyphenols, and LAB [10]. Fermentation is initiated by various endogenous microorganisms derived from the raw materials added to kimchi, such as *Lactobacillus* spp., *Pediococcus* spp., *Leuconostoc* spp., and *Enterococcus* spp. LAB become the predominant species during incipient fermentation, producing lactic acid and decreasing the pH of foods [11]. During the storage of kimchi, these bacteria produce the unique flavor of kimchi and also produce compounds that are beneficial to health through fermentation [12]. Rho et al. [13] isolated the probiotic *E. faecium* FC-K with anti-allergic activity by modulating the type 2 T-helper (Th2)-mediated pathological response from kimchi. Valledor et al. [14] reported the antimicrobial activity of the two enterocin-producing *E. faecium* ST20Kc and ST41Kc. Ahn et al. [15] reported the immunomodulatory activity probiotic *E. faecium* JS1-8.

Therefore, our study confirmed the safety of two *Enterococcus* strains, *E. faecalis* MG5206 and *E. faecium* MG5232, isolated from kimchi, through an assessment of antibiotic resistance and various virulent genes, and we also evaluated their potential as probiotics for use in the food industry.

## 2. Materials and Methods

### 2.1. Bacterial Strains and Culture Conditions

The *E. faecalis* MG5206 (MG5206) and *E. faecium* MG5232 (MG5232) used in this study were previously isolated from kimchi. Strain isolation was performed according to previously described methods [16]. The bacteria were cultured in de Man Rogosa Sharpe (MRS) medium (Difco, Detroit, MI, USA) at 37 °C for 24 h. Isolated LAB were identified by 16s ribosomal RNA (rRNA) sequencing, prepared with 25% glycerol stock, and stored at −70 °C until the experiment.

### 2.2. Whole-Genome Sequencing Analysis (WGS)

The genomic DNA of MG5206 and MG5232 was extracted using the PureLink™ Microbiome DNA Purification Kit (Thermo Fisher Scientific Inc., Waltham, MA, USA). Genome sequencing was performed by Macrogen Inc. (Seoul, Korea) using a PacBio RS II instrument (Pacific Biosciences of California Inc., Menlo Park, CA, USA) on an Illumina platform (Illumina Inc., San Diego, CA, USA). The data were assembled using a Hierarchical Genome Assembly Process (HGAP3). After the assembly process, Illumina reads were applied for accurate genome sequencing using Pilon (version 1.21; Broad Institute, Cambridge, MA, USA). To validate the accuracy of the assembly, the Illumina reads were mapped to the assembly results. Based on NCBI data and BLAST analysis, we identified the species to which each scaffold showed similarity. The assembled gene was analyzed for sequence homology and annotated based on the Prokka (version 1.12b) and EggNOG (version 4.5) databases to estimate information on coding sequences, transfer RNA (tRNA), and rRNA gene information [17,18].

### 2.3. In Vitro Safety Test of the Strains MG5206 and MG5232

#### 2.3.1. Antibiotic Susceptibility

An antibiotic resistance of the strains MG5206 and MG5232 was assayed using the minimum inhibitory concentration (MIC) test strip method. The pellets of each strain were harvested by centrifugation (3460× *g*, 10 min) after culturing in MRS broth at 37 °C for 18 h, washed twice with phosphate-buffered saline (PBS, pH 7), and resuspended in PBS to obtain a resuspended solution with a McFarland turbidity level of 0.5. The suspended solution was inoculated in a lactic acid bacteria susceptibility test medium (LSM) plate, with a mixture of 90% Iso-Sensitest medium (Oxoid Ltd., Hampshire, UK) and 10% MRS with 1.5% agar, using swabs [19]. The plates were dried for 10 min, and MIC test strips (Liofilchem Inc., Roseto degli Abruzzi, Italy) were placed on the plate according to the manufacturer’s instructions and incubated at 37 °C, and the results were recorded 24 h after inoculation. Antibiotic susceptibility was determined according to the European Food Safety Authority (EFSA) guidelines [20].

#### 2.3.2. Hyaluronidase Activity

Hyaluronidase activity was determined according to the method described by Hynes et al. [21], with slight modifications. Hyaluronic acid (HA) and bovine albumin fraction V (BAFV) were added up to 400 µg/mL and 1%, respectively, in brain heart infusion (BHI) medium with 1% agar. The BHI-HA-BAFV medium was allowed to solidify in petri dishes. One drop of bacteria, cultured in MRS broth, was inoculated onto the BHI-HA-BAFV agar plate and incubated at 37 °C for 24 h. After 10 min of staining with 2N acetic acid, hyaluronidase activity was detected as a zone of clearance around the bacterial colonies. No activity was determined with a cloudy background appearance by acetic acid precipitation of an albumin and non-degraded hyaluronic acid complex. *Staphylococcus aureus* KCCM12214 was used as a positive control.

#### 2.3.3. Gelatinase Activity

Gelatinase activity was determined by gelatin liquefaction according to previously described methods, with certain modifications [22]. MG5206 and MG5232 cultures, grown in MRS broth for 18 h at 37 °C, were inoculated into gelatin nutrient medium (MRS with 0.12 g/mL gelatin, 0.005 g/mL peptone, and 0.03 g/mL beef extract). The inoculated medium was incubated for 9 days at 37 °C. The cultured gelatin nutrient medium was refrigerated for 4 h at 4 °C, and solidification was observed by slant formation.

#### 2.3.4. Hemolytic Activity

To evaluate the hemolytic activity, MG5206 and MG5232 were grown in MRS broth for 18 h and streaked onto tryptic soy agar (Oxoid Ltd., Hampshire, UK) with 5% sheep blood (MB cell, Seoul, Korea). After incubation at 37 °C for 24 h under aerobic conditions, hemolytic properties were evaluated based on the lysis of red blood cells around the colonies [23].

### 2.4. Acute Toxicity Study

Acute toxicity tests were performed on Sprague–Dawley (SD) rats at Chemon Inc. (Yongin, Korea) and approved by the Animal Ethics Committee of the Institutional Animal Care and Use Committee of Chemon Inc. (No. 20-RA-0146, 0147). The acute oral toxicity study was performed according to the protocol of the Organization for Economic Co-operation and Development (OECD) Guidelines No. 420 for “Acute Oral Toxicity-Fixed Dose Procedure” [24]. Following a 1 week period of adaptation, 70 healthy SD rats (5 rats of each sex in each group, 7 weeks of age) were randomly divided into 7 groups. The animals were caged individually in rooms with controlled temperature and humidity (22 ± 2 °C with 50 ± 5% humidity) and with a 12 h light-dark cycle. Food and water were provided ad libitum to the animals. After 16 h of fasting, MG5206 or MG5232 (1.4 × 10^10^ CFU/g in distilled saline solution) was orally administered to animals in the test groups at a concentration of 1250 (low dose), 2500 (middle dose), and 5000 mg/kg body weight (high dose). The animals in the control group were administered saline solution without any bacterial strains. Following a single oral administration of the strains, the clinical signs of the animals were monitored for 14 days. The body weight and food intake of the rats were measured once every 2 days. On day 15, all the rats were fasted for 12 h and anesthetized using CO_2_. Necropsy was performed on all the rats by visual inspection. All the histological evaluations were performed by a board-certified toxicological pathologist.

### 2.5. In Vitro Characterization of the Strains MG5206 and MG5232

#### 2.5.1. Autoaggregation

Autoaggregation assays were performed according to previously described methods, with certain modifications [25]. The bacteria were grown for 20 h at 37 °C in MRS broth. The cells were harvested by centrifugation, washed twice, and resuspended in PBS (pH 7.2). Suspensions (4 mL) were mixed by vertexing for 10 s. Then, 0.1 mL of the upper suspension was mixed with 0.9 mL PBS, and the absorbance (A0) at 600 nm was measured. Using the same method, the absorbance (A1) was determined after 5 h of incubation at room temperature. The autoaggregation percentage was expressed as (1 − A1/A0) × 100.

#### 2.5.2. Bacterial Adhesion to Solvents

The bacterial adhesion to solvents (BATS) assay was performed using the method described by Kos et al. [25] with modifications. The bacterial cells were suspended in PBS (pH 7.2) at a concentration of 1 × 10^8^ CFU/mL. The cell suspension (3 mL) was mixed with 1 mL of solvent. Xylene was used as the apolar solvent, chloroform as the electron acceptor, and ethyl acetate as the electron donor. The mixture was vortexed for 1 min and allowed to stand for 20 min to separate into two phases. Absorbance of the aqueous phase was measured at 600 nm using a UV spectrophotometer. The affinities to solvents with different physicochemical properties (hydrophobicity and electron donor–electron acceptor interactions) were expressed using the following equation:BATS (%) = (1 − At/A0) × 100
where A0 and At are the absorbances before and after extraction with the organic solvents, respectively.

#### 2.5.3. Survivability in Simulated Gastrointestinal Condition

To evaluate the gastrointestinal resistance, the strain was harvested by centrifugation (3460× *g*, 10 min) after culturing in MRS broth at 37 °C for 24 h. The pellets were washed twice with PBS (pH 7) and resuspended (10^8^ CFU/mL) in simulated gastric fluid (SGF; 3 g/L of pepsin in PBS, pH 3 and pH 4) and simulated intestinal fluid (SIF; 1 g/L of pancreatin in PBS, pH 7 and 8). Viable cells were determined on MRS agar after incubation at 37 °C for 3–4 h [26]. Bacterial survival rate was calculated using the following equation:(1)$$\mathrm{Survival}\;\left(\%\right)=\frac{\mathrm{Log}\mathrm{CFU}\;\mathrm{of}\;\mathrm{viable}\;\mathrm{cells}\;\mathrm{survived}}{\mathrm{Log}\mathrm{CFU}\;\mathrm{of}\;\mathrm{initial}\;\mathrm{viable}\;\mathrm{cells}}\times 100$$

#### 2.5.4. Enzyme Production and Carbohydrate Fermentation

Enzyme activities and carbohydrate fermentation of the strains were assessed using an API ZYM and API 50 CHL kit (Bio-Merieux, Lyon, France) according to the manufacturer’s instructions. Enzyme and fermentation activities were evaluated by color change, and color intensity was compared with the color chart provided by the manufacturer.

## 3. Results and Discussion

### 3.1. Whole-Genome Sequencing (WGS) Analysis of MG5206 and MG5232

WGS analysis is a highly discriminatory technique used to investigate the biological and evolutionary characteristics of bacterial species. The derived data allow for the identification of phylogenetic relationships between species, obtaining molecular markers for the strains, and performing comparative analyses to investigate drug-resistant genes [26]. The virulence of enterococci is due to the presence of their virulence factors and resistance to various antibiotics [27]. Therefore, the genomic characterization of *Enterococcus* spp. can be a crucial strategy for counteracting bacterial infections and their utilization as probiotics.

In our study, DNA extraction and WGS analyses were performed to identify the antibiotic resistance and virulence genes of the strains MG5206 and MG5232. After genome assembly, we analyzed the sequence homology based on databases. The assembled genome of MG5206 consisted of 2,764,656 base pairs (bp) with a GC content of 37.65%. The MG5232 genome has two contigs and consists of 2,777,286 bp with a GC content of 37.92% (Table 1). In addition, genetic maps of the circular genomes of the strains MG5232 and MG5206 are presented in Figure 1.

A comparative analysis was performed using two antibiotic resistance gene databases (CARD and ResFinder) to identify the resistance genes for 10 antibiotics (Ampicillin, Chloramphenicol, Clindamycin, Erythromycin, Gentamycin, Kanamycin, Streptomycin, Tetracycline, Tylosin, and Vancomycin) of MG5206 and MG5232. It was confirmed that aminoglycoside antibiotic resistance genes (*aac(6′)-**I, ant(3’)-III, ant(6’)-Ia, and aph(2’)-Id*), as well as macrolide antibiotic resistance genes containing erythromycin and tylosin ((*eatAv and msr(C)*), did not exist [28,29].

In addition, nine types of vancomycin resistance Enterococci have been reported, and *vanA* and *vanB* account for most of the global prevalence [30]. In this study, MG5206 and MG5232 did not contain any vancomycin resistance genes. Therefore, we confirmed that there was no possibility of pathogen transfer.

The virulence factors of *Enterococcus* include cytolysin, aggregation substances, gelatinase, enterococcal surface proteins, and hyaluronidase. Cytolysin (*cylA, cylB, cylI, cylL-l, cylL-s, cylM, cylR1, and cylR2*) produces β-hemolysin-bacteriocin; the aggregation substance (*asa1, asp1, agg*) attaches to the surface of eukaryotic cells and penetrates into the cell; enterococcal surface protein (*esp*) is involved in colonization and maintenance of the urinary tract and cardiac catheters and plays a role in biofilm formation; gelatinase (*gelE*) hydrolyzes collagen and gelatin, exacerbates endocarditis, and participates in biofilm formation; hyaluronidase (*hyl*) affects invasive diseases [31,32]. In our study, the MG5206 and MG5232 genomes did not contain these virulence factors.

Therefore, from genomic analysis, it can be concluded that the strains MG5206 and MG5232 do not harbor any virulence genes and have no transmissible antibiotic resistance or virulence factors.

Functional analyses of the representative genes in the core genomes were performed using the Prokka software tool (version 1.12b) (Table 2). A total of 2553 coding sequences (CDSs) of MG5206 were annotated as functional core genes, wherein 862 CDSs (33.8%) were involved in translation, ribosomal structure, biogenesis, transcription, carbohydrate transport and metabolism, amino acid transport and metabolism, replication, recombination, and repair. A total of 673 unknown genes (26.3%) are listed.

Of the 2582 CDSs of MG5232, 1096 (42.4%) were annotated as functional core genes involved in translation, ribosomal structure, replication, recombination, repair, carbohydrate transport and metabolism, amino acid transport and metabolism, and cell wall/membrane/envelope biogenesis, and 631 unknown genes (24.4%) are listed.

### 3.2. Identification of Phenotypic Characteristics of the Strains MG5206 and MG5232

#### 3.2.1. Antibiotic Resistance

The transmission of antibiotic resistance genes into pathogens due to the abuse of broad-spectrum antibiotics is one of the safety issues in probiotic strains [33]. *Enterococcus* spp. are an opportunistic pathogen that causes serious infections and has been reported to be multidrug-resistant (MDR) to various antibiotics, including vancomycin [34]. Enterococci are naturally and intrinsically resistant to various antibacterial agents such as penicillin, ampicillin, and most cephalosporins [35].

In our study, the MICs of 10 antibiotics were evaluated to ensure the safety of the strains MG5206 and MG5232. The MICs of all 10 antibiotics for MG5206 and MG5232 were equal to or lower than the cut-off MIC values of the EFSA guidelines and were therefore identified as susceptible (Table 3). In view of this, MG5206 and MG5232 were classified as usable probiotics according to the EFSA guidelines.

#### 3.2.2. Hyaluronidase and Gelatinase Activity

Some enzymes, such as gelatinase and hyaluronidase, are related to virulence traits that establish and spread infections because of their catalytic activity in the invasion of host tissues [36,37]. Moreover, the gelatinase and hyaluronidase activities of probiotic bacteria may cause serial infections in the host. Therefore, their activities should be assessed in order to exclude potentially harmful probiotic candidates.

Hyaluronidase is a glycosidase enzyme that mainly degrades the hyaluronic acid, a mucopeptide composed of alternating residues of N-acetyl glucosamine, and glucuronic acid [21,38]. However, various pathogenic organisms produce hyaluronidases, including streptococci, pneumococci, staphylococci, and clostridia [21]. As a mentioned earlier, hyaluronidase as a virulent factor affects invasive diseases.

Gelatinases, as a matrix metalloproteinase (MMPs), degrade almost all of the extracellular matrix (ECM), including basement membrane components, and might provide a suitable substrate for further activity of human gelatinases or other bacterial proteinases [22,39]. Gelatinase is mostly produced by pathogenic microorganisms and biofilm composites [38]. As a virulence factor, it exerts toxicity on the host by decomposing various types of proteins, such as collagen, fibrinogen, and complement, which are involved in immunity. It is also known to cause problems, such as disturbance of the host immune system and induction of endocarditis, mainly because of its extensive proteolytic activity [40,41].

In our study, the strains MG5206 and MG5232 were confirmed to not have hyaluronidase and gelatinase activity (Table 4).

#### 3.2.3. Hemolytic Activity

Hemolytic activity refers to the ability of a pathogen or disease to destroy red blood cells [23]. In addition, evaluation of the hemolytic activity of probiotics using in food products is strongly recommended, even if they are generally recognized as safe (GRAS) or quality presumption of safety (QPS) [20]. Hemolysis caused by bacteria can be divided into α-, β-, and γ-hemolysis [42]. α-hemolysis is the partial lysis of red blood cells and hemoglobin that changes the colony area to green after incubation. γ-hemolysis does not cause hemolysis. In contrast, in β-hemolysis, red blood cells and hemoglobin are entirely dissolved, and the colony becomes transparent after incubation. *Enterococcus* spp., producing a β-hemolysin-bacteriocin called cytolysin, have been reported to increase the infection rate by five times in patients [43]. According to Deng’s study [44], among 110 probiotic *Enterococcus* spp., 35 (31.8%) showed β-hemolysis. In Sanlibaba’s study [45], of 97 *Enterococcus* spp., 12.37% of enterococcal strains showed β-hemolytic characteristics. In our study, the strains MG5206 and MG5232 showed γ-hemolysis. Neither strain showed β-hemolytic ability (Figure 2).

Based on these results, MG5206 and MG5232 is inferred to not possess antibiotic resistance, hyaluronidase, gelatinase, or hemolytic activities. Therefore, in terms of safety, it was confirmed that the strains MG5206 and MG5232 were suitable for use as probiotics.

### 3.3. Acute Toxicity

*Enterococcus* spp. can cause gastroenteritis, endocarditis, UTIs, and meningitis in people with weakened immunity [46]. The acute oral toxicity test is fundamental for evaluating the safety of probiotic bacterial strains [24,47].

In our study, the safety of the strains MG5206 and MG5232 was evaluated using acute oral toxicity tests in SD rats. At the end of the test period, all animals survived, appeared healthy, and showed normal growth and development patterns. As shown in Figure 3, oral administration of MG5206 or MG5232 did not affect the survival rate, general symptoms, body weight, urinalysis, or gross necropsy findings in experimental rats. Therefore, data recorded in the present experimental study indicated that the approximate lethal dose (ALD) after a single oral administration of MG5206 or MG5232 in both sexes of SD rats was considered to be more than 5000 mg/kg body weight.

### 3.4. Probiotic Properties of the Strains MG5206 and MG5232

#### 3.4.1. Autoaggregation Ability

The autoaggregation ability of probiotics was used to indirectly evaluate the degree of intestinal adhesion in vitro. Autoaggregation is strain-specific and may vary within the same taxonomic group [48]. The ability of probiotics to adhere to the intestinal mucosa can contribute to the discharge of harmful bacteria into the intestine by inhibiting the adhesion of pathogenic microorganisms to the intestinal mucosa and preventing colony formation. In contrast, high autoaggregation ability in pathogens is considered a virulence factor because it forms colonies on the intestinal mucosa and increases antibiotic resistance [49]. Compared to previous studies, the autoaggregation ability of various probiotic strains was approximately 30–96%, with an average of 62.6% [16], and in particular, *E. faecium* MG89-2 showed approximately 60–70% autoaggregation [16]. Our study confirmed that MG5206 and MG5232 showed a lower autoaggregation ability than general LAB. In our study, the autoaggregation ability of MG5206 and MG5232 was between 37.2% and 66.3%, and the autoaggregation ability of MG5232 alone was approximately half of that of MG5232 (Table 5). In Mansour’s study, *E. faecium* NM1015 showed a higher autoaggregation ability than *E. faecalis* NM815 or NM915 [50]. Therefore, this difference in autoaggregation between MG5206 and MG5232 was also considered a characteristic of the strain.

No virulence genes were related to aggregations in the WGS of MG5206 and MG5232, and the actual autoaggregation ability was not high. Based on these results, both the strains are considered safe and do not show toxicity in the body.

#### 3.4.2. Hydrophobicity

Cell surface hydrophobicity is a non-specific interaction between microbial cells and their hosts [51]. The initial interaction may be weak, often reversible, and precedes subsequent adhesion processes mediated by more specific mechanisms involving cell-surface proteins and lipoteichoic acids. Bacterial cells with high hydrophobicity usually exhibit strong interactions with mucosal cells. Hydrophobicity may assist in adhesion but is not a prerequisite for strong adherence to host cells. Furthermore, hydrophobicity varies among close genetically related species and strains of the same species [52].

In our study, MG5206 showed a low level of hydrophobicity (11.90 ± 1.66%), as determined by adhesion to ethyl acetate. Low hydrophobicity was also observed in MG5232 (12.69 ± 1.18%, 28.88 ± 1.95%), as determined by adhesion to xylene and chloroform, respectively (Table 5).

#### 3.4.3. Survival under Conditions Simulating the Human Gastrointestinal Tract

The low pH of the stomach and antimicrobial action of pepsin are practical barriers to the entry of bacteria into the gastrointestinal tract. Therefore, tolerance of humans to gastric pH is an important factor in the selection of probiotics.

Gastric transit was observed from 0 to 90 min and then up to 180 min [53]. In the present study, we observed that MG5206 and MG5232 survived appreciably at pH 3 and 4 (Table 6). In addition, both the strains survived in pancreatin conditions at pH 7 and 8 (Table 6). Tolerance to bile salts and pancreatin is essential for LAB survival in the small intestine [54]. These results confirmed that MG5206 and MG5232 are likely to survive in the stomach and intestine.

#### 3.4.4. Characterization of Enzyme Production and Carbohydrate Fermentation

The enzymes produced by microorganisms must be evaluated to identify potentially toxic substances in humans [11]. In our study, MG5206 and MG5232 did not produce alkaline phosphatase, lipase, trypsin, α/β-galactosidase, β-glucuronidase, α/β-glucosidase, N-acetyl-β-glucosaminidase, α-mannosidase, or α-fucosidase (Table 7). The LAB-producing β-glucuronidase is a carcinogenic enzyme that adversely affects the liver [55]. In contrast, esterase, esterase lipase, leucine arylamidase, acid phosphate, and naphthol-AS-BI-phosphohydrolase were produced by the strains MG5206 and MG5232. Valine arylamidase and crystine arylamidase are produced only by MG5232, and α-chymotrypsin is produced only by MG5206.

In addition, both the strains MG5206 and MG5232 were capable of fermenting different sugars, including D-glucose, D-fructose, D-mannose, D-maltose, and D-sucrose (Table 8). It is expected that the sugar usage capacity confirmed in our study can be utilized to increase the productivity of these strains.

## 4. Conclusions

*Enterococcus* spp. are used in food fermentation because of their advantageous characteristics, such as stability and fermentability. However, since the toxicity of *Enterococcus* spp. has been considered, safety issues on potential antibiotic resistance and various virulence factors continue to arise. Therefore, it is essential to prove the safety of the newly discovered *Enterococcus* spp. The absence of virulence genes or factors for MG5206 and MG5232 in this study was confirmed from WGS data. They also showed no antibiotic resistance, hemolytic activity, or acute oral toxicity. In addition, the high gastrointestinal survival rate of the two strains suggested that these strains could be utilized in food as potential probiotics in the future. Moreover, confirmation of the enzyme activity and carbohydrate fermentation ability of these two strains is expected to be utilized in future research to improve productivity. Therefore, *E. faecalis* MG5206 and *E. faecium* MG5232 with secured safety and stability are expected to be used as probiotics in various food fields in the future.

## Figures and Tables

**Figure 1 microorganisms-10-02070-f001:**
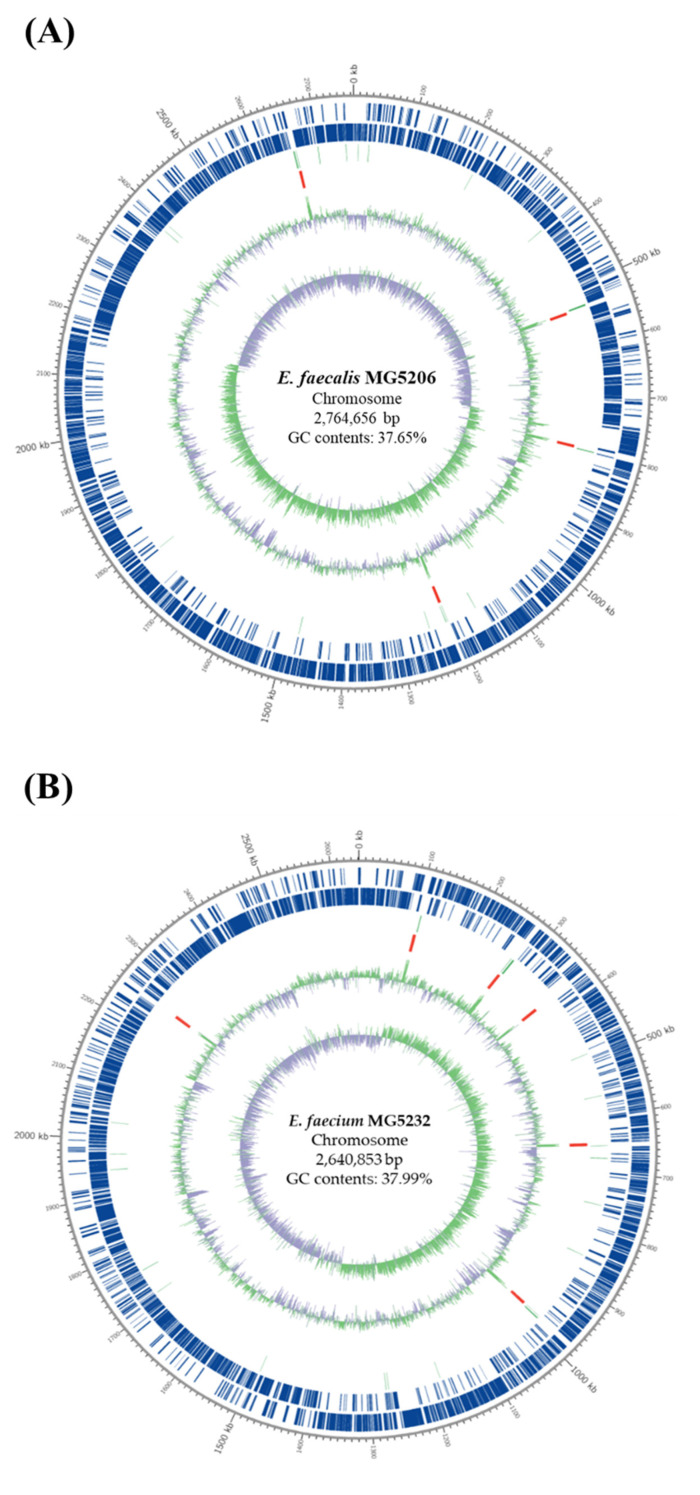
Genomic map of *E. faecalis* MG5206 and *E. faecium* MG5232. Marked genome characteristics are shown from outside to the center. (**A**) The characteristic of *E. faecalis* MG5206 chromosomal DNA; (**B**) the characteristic of *E. faecium* MG5232 chromosomal DNA; (**C**) the characteristic of *E. faecium* MG5232 plasmid; coding sequence (CDS) on the forward strand, CDS on the reverse strand, tRNA, rRNA, GC content, and GC skew. The region of tRNA is marked in light green and that of rRNA is marked in red. The exterior light green peak describes the region with a higher value of GC percentage than average. Otherwise, it is described in the interior marked as lavender peak; the height of the peak describes the difference from the average GC percentage. According to the formula (G − C)/(G + C), a positive value shows that G is dominant, while a negative value shows that C is dominant. The exterior light green peak describes the region with higher G content, while the interior lavender peak describes the region with higher C content.

**Figure 2 microorganisms-10-02070-f002:**
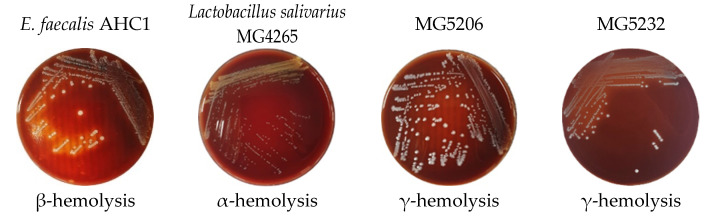
Phenotypic hemolysis of in *E. faecalis* MG5206 and *E. faecium* MG5232. β-hemolysis control, *E. faecalis* AHCI; α-hemolysis control, *Lactobacillus salivarius*.

**Figure 3 microorganisms-10-02070-f003:**
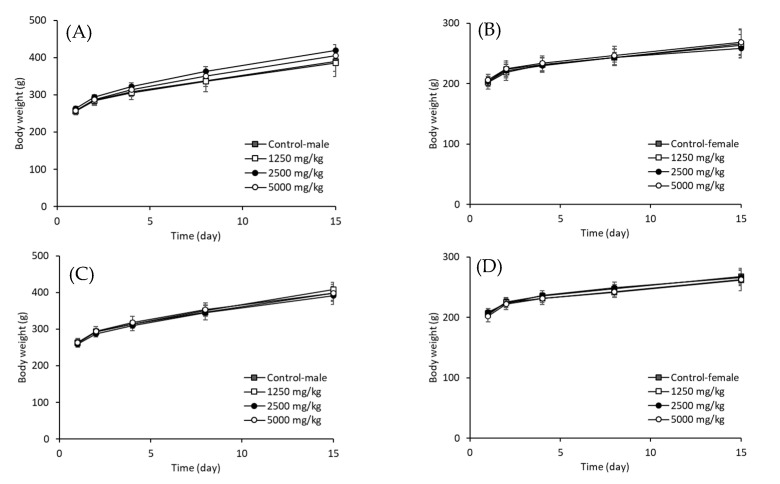
Changes in the body weight of SD rats for 14 days after single-dose administration of *E. faecalis* MG5206 and *E. faecium* MG5232 in the acute toxicity study. MG5206 or MG5232 (1.4 × 10^10^ CFU/g in distilled saline solution) was orally administered to at a concentration of 1250, 2500, and 5000 mg/kg body weight, respectively. Body weight of MG5206 administered to (**A**) male and (**B**) female rats; body weight of MG5232 administered to (**C**) male and (**D**) female rats. The data are presented as the means ± SD (*n* = 5).

**Table 1 microorganisms-10-02070-t001:** General genomic information of *E. faecalis* MG5206 and *E. faecium* MG5232.

Feature	MG5206	MG5232
No. of contigs	1	2
Chromosome size (bp)	2,764,656	2,640,853
Plasmid size (bp)	-	136,433
G + C contents (%)	37.65	37.92
Total genes	2635	2707
rRNA genes	12	18
tRNA genes	61	68
Coding sequence (CDS)	2562	2621
Total genome size (bp)	2,764,656	2,777,286
No. of plasmid	0	1

**Table 2 microorganisms-10-02070-t002:** Functional categories of core genes in *E. faecalis* MG5206 and *E. faecium* MG5232 genomes.

EggNOG	Function	MG5206		MG5232	
		No. of CDS	Ration of CDS	No. of CDS	Ration of CDS
J	Translation, ribosomal structure, and biogenesis	157	6.150	148	5.732
A	RNA processing and modification	0	0.000	0	0.000
K	Transcription	191	7.481	190	7.358
L	Replication, recombination, and repair	123	4.818	193	7.475
B	Chromatin structure and dynamics	0	0	0	0.000
D	Cell cycle control, cell division, and chromosome partitioning	20	0.783	20	0.775
Y	Nuclear structure	0	0	0	0.000
V	Defense mechanisms	59	2.311	53	2.053
T	Signal transduction mechanisms	62	2.429	65	2.517
M	Cell wall/membrane/envelope biogenesis	121	4.740	133	5.151
C	Cell motility	5	0.196	4	0.155
Z	Cytoskeleton	0	0.000	0	0.000
W	Extracellular structures	0	0.000	0	0.000
U	Intracellular trafficking, secretion, and vesicular transport	23	0.901	27	1.046
O	Posttranslational modification, protein turnover, and chaperones	56	2.194	59	2.285
C	Energy production and conversion	95	3.721	73	2.827
G	Carbohydrate transport and metabolism	218	8.539	286	11.077
E	Amino acid transport and metabolism	173	6.776	146	5.655
F	Nucleotide transport and metabolism	89	3.486	73	2.827
H	Coenzyme transport and metabolism	57	2.233	40	1.549
I	Lipid transport and metabolism	52	2.037	49	1.898
P	Inorganic ion transport and metabolism	131	5.131	109	4.222
Q	Secondary metabolites biosynthesis, transport, and catabolism	17	0.666	15	0.581
R	General function prediction only	231	9.048	268	10.380
S	Function unknown	673	26.361	631	24.438
	Total	2553	100	2,582	100

CDS, coding sequence.

**Table 3 microorganisms-10-02070-t003:** Minimum inhibitory concentration (MIC) and antibiotic susceptibility of *E. faecalis* MG5206 and *E. faecium* MG5232.

Antibiotics	MIC (µL/mL)		Cut-Off Value (µL/mL)
	MG5206	MG5232	
Ampicillin	2.0 ± 0.0	2.0 ± 0.0	2.0
Chloramphenicol	8.0 ± 0.0	8.0 ± 0.0	16.0
Clindamycin	4.0 ± 0.0	2.0 ± 0.0	4.0
Erythromycin	2.0 ± 0.0	4.0 ± 0.0	4.0
Gentamycin	32.0 ± 0.0	8.0 ± 0.0	32.0
Kanamycin	256.0 ± 0.0	1024.0 ± 0.0	1024.0
Streptomycin	128.0 ± 0.0	64.0 ± 0.0	128.0
Tetracycline	2.0 ± 0.0	4.0 ± 0.0	4.0
Tylosin	4.0 ± 0.0	4.0 ± 0.0	4.0
Vancomycin	2.0 ± 0.0	2.0 ± 0.0	4.0

MIC, minimum inhibitory concentration; microbiological cut-off values for antibiotics for *Enterococcus* spp., as provided by EFSA guidelines [20].

**Table 4 microorganisms-10-02070-t004:** Phenotypic characteristics of *E. faecalis* MG5206 and *E. faecium* MG5232.

Strains	Hyaluronidase	Gelatinase
MG5206	(-)	(-)
MG5232	(-)	(-)

(-), no activity.

**Table 5 microorganisms-10-02070-t005:** Autoaggregation ability and hydrophobicity of the strains *E. faecalis* MG5206 and *E. faecium* MG5232.

Strains	Autoaggregation (%)	Adhesion to Solvents (%)		
		Xylene	Chloroform	Ethyl Acetate
MG5206	37.20 ± 3.40	65.31 ± 1.01	94.89 ± 1.73	11.90 ± 1.66
MG5232	66.30 ± 1.30	12.69 ± 1.18	28.88 ± 1.95	54.60 ± 15.73

**Table 6 microorganisms-10-02070-t006:** Survival of the *E. faecalis* MG5206 and *E. faecium* MG5232 in simulated human gastrointestinal tract conditions.

Strains	Viable Counts (log CFU/mL)			
	Simulated Gastric Fluid ^1^		Simulated Intestinal Fluid ^2^	
	pH 3	pH 4	pH 7	pH 8
MG5206	4.79 ± 0.01	7.84 ± 0.03	7.92 ± 0.01	7.90 ± 0.02
MG5232	6.93 ± 0.04	7.62 ± 0.06	7.56 ± 0.04	7.58 ± 0.02

^1^ Simulated gastric tolerance is shown as viable counts (log CFU/mL) at pH 3 and pH 4 after a 3 h reaction. ^2^ Simulated intestinal tolerances are shown as viable counts (log CFU/mL) at 37 °C after 4 h of the reaction.

**Table 7 microorganisms-10-02070-t007:** Enzyme activities of *E.* MG5206 and *E. faecium* MG5232 using API ZYM kit.

Enzymes	Substrate	MG5206	MG5232
Alkaline phosphatase	2-naphthyl phosphate	0	0
Esterase (C4)	2-naphthyl butyrate	3	3
Esterase Lipase (C8)	2-naphthyl caprylate	2	2
Lipase (C14)	2-naphthyl myristate	0	0
Leucine arylamidase	L- leucyl-2-naphthylamide	3	3
Valine arylamidase	L-valyl-2-naphthylamide	0	1
Crystine arylamidase	L-cystyl-2-naphthylamide	0	2
Trypsin	N-benzoyl-DL-arginine-2-naphthylamide	0	0
α-chymotrypsin	N-glutaryl-phenylanine-2-naphthylamide	1	0
Acid phosphatase	2-naphtyl phosphate	2	2
Naphthol-AS-BI-phosphohydrolase	Naphthol-AS-BI-phosphate	3	2
α-galactosidase	6-Br-2-naphthyl-*α*D-galactopyranoside	0	0
β- galactosidase	2-naphthyl-*β*D-galactopyranoside	0	0
β-glucuronidase	Naphthol-AS-BI-*β*D-glucuronide	0	0
α-glucosidase	2-naphthyl-*α*D-glucopyranoside	0	0
β-glucosidase	6-Br-2-naphthyl- *β*D-glucopyranoside	0	0
N-acetyl-β-glucosaminidase	1-naphthyl-N-acetyl-*β*D-glucosaminide	0	0
α-mannosidase	6-Br-2-naphthyl-*α*D-mannopyranoside	0	0
α-fucosidase	2-naphthyl-*α*L-fucopyranoside	0	0

**Table 8 microorganisms-10-02070-t008:** Carbohydrate fermentation characteristics of *E. faecalis* MG5206 and *E. faecium* MG5232 using API 50 CHL kit.

Substrate	MG5206	MG5232	Substrate	MG5206	MG5232
Glycerol	+ ^1^	+	Salicin	+	+
Erythritol	− ^2^	−	D-cellobiose	+	+
D-arabinose	−	−	D-maltose	+	+
L-arabinose	+	+	D-lactose	+	+
D-ribose	+	+	D-melibiose	−	−
D-xylose	+	+	D-sucrose	+	+
L-xylose	−	−	D-trehalose	+	+
D-adonitol	−	−	Inulin	−	−
Methyl-β D-xylopyranoside	−	−	D-melezitose	+	−
D-galactose	+	+	D-raffinose	−	−
D-glucose	+	+	Starch	+	−
D-fructose	+	+	Glycogen	−	−
D-mannose	+	+	Xylitol	−	−
L-sorbose	−	−	Gentiobiose	+	+
L-rhamnose	+	−	D-turanose	−	−
Dulcitol	−	−	D-lyxose	−	−
Inositol	−	−	D-tagatose	+	−
D-mannitol	+	+	D-fucose	−	−
D-sorbitol	+	+	L-fucose	−	−
Methyl-α D-mannoside	+	+	D-arabitol	−	−
Methyl-α D-glucoside	−	−	L-arabitol	−	−
N-acetyl-glucosamine	+	+	Gluconate	+	−
Amygdalin	+	+	2-keto-gluconate	−	−
Arbutin	+	+	5-keto-gluconate	−	−
Esculin	+	+			

^1^ + means that carbohydrates can be fermented. ^2^ – means that carbohydrates cannot be fermented.

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
