# Peer review of "Evaluation of Safety and Probiotic Potential of Enterococcus faecalis MG5206 and Enterococcus faecium MG5232 Isolated from Kimchi, a Korean Fermented Cabbage"

_microorganisms, 2022, doi:10.3390/microorganisms10102070_

Round 1
Reviewer 1 Report
The authors provided a complete description about two strains of potential probiotics properties. The authors performed whole genome sequencing of the different strains and provided the necessary genetic information such as GC content and genome size, enzymes, virulence factors. They tested the different phenotypic activities of the two strains, and they concluded that they do not have hemolytic activity, or virulence factors, and antibiotic resistance. The authors made it easy to describe the methodology.
Comments
1. Did the authors inject the two strains in immunocompromised mouse model such as SCID mice to have a clear picture about pathogenicity instead of the tested rat model.
2. Minor mistakes were highlighted.
3. the authors should elaborate on the future directions of the study.
Author Response
Reviewer 1
Thanks for your attentive comments.
Comments
- Did the authors inject the two strains in immunocompromised mouse model such as SCID mice to have a clear picture about pathogenicity instead of the tested rat model.
Reply: The two strains used in this study are will be utilized as probiotics in food field through further studies. Therefore, we confirmed safety using only healthy SD rats in this study.
- Minor mistakes were highlighted.
Reply: Thank you for your comment. However, “the highlighted” part could not be confirmed. We corrected typos and formatting throughout the manuscript and marked them in red.
- the authors should elaborate on the future directions of the study.
Reply: Accordance with reviewer’s comment, we provided future directions for strain utilization in lines 410-411.

Reviewer 2 Report
This study have confirmed the safety of two Enterococcus strains, Enterococcus faecalis MG5206 and Enterococcus faecium MG5232, isolated from kimchi, through an assessment of antibiotic resistance and various virulent genes after Whole-genome sequencing analysis. The authors confirmed there is no virulence genes or factors, antibiotic resistance, hemolytic activity, or acute oral toxicity in these strains. Besides, the high gastrointestinal survival rate of the two strains suggested their potentiality as probiotics in the future. In general, this study is very interesting, well designed, presented and discussed.
Minor corrections
1- Highlight the data of Whole-genome sequencing analysis of both strains at abstract and conclusion section.
2- Correct formatting of Enterococcus at L20
3- Expand WGS at L175
4- Figure 1: clarify as a,b, and c for the 3 presented figures on both fig and legend
5- L394, add using API 50 CHL kit
6- Confirm the formatting of microorganisms at all reference section see for example L428,433, 437, 446, 448, 476,, 479, 480, ……etc
Author Response
Reviewer 2
Thanks for your attentive comments.
Comments
- Highlight the data of Whole-genome sequencing analysis of both strains at abstract and conclusion section.
Reply: Accordance with reviewer’s comment, we supplemented the WSG content in the sentences on lines 12-13 and 404-405 of the manuscript.
- Correct formatting of Enterococcus at L20
Reply: We corrected Enterococcus in the keywords part of L21.
- Expand WGS at L175
Reply: We modified the “WSG” in L177 to “Whole-Genome Sequencing (WSG)”.
- Figure 1: clarify as a,b, and c for the 3 presented figures on both fig and legend
Reply: Accordance with reviewer’s comment, we marked (A), (B), and (C) in the Figure 1, and corrected the figure legend in lines 230-233.
- L394, add using API 50 CHL kit
Reply: We added the phrases “API 50 CHL kit” and “API ZYM kit” to Table 7 and Table 8 on lines 395 and 398, respectively.
- Confirm the formatting of microorganisms at all reference section see for example L428,433, 437, 446, 448, 476,, 479, 480, ……etc
Reply: Accordance with reviewer’s comment, we corrected the formatting of all microorganisms in the Reference section of the manuscript.
